# *firebehavioR*: An R Package for Fire Behavior and Danger Analysis

**Justin P. Ziegler [1,\*], Chad M. Hoffman [1] and William Mell [2]**

1   Forest and Rangeland Stewardship Department, Colorado State University, Fort Collins, CO 80523, USA
2   Pacific Northwest Research Station, Forest Service, U.S. Department of Agriculture, Seattle, WA 87545, USA
\*   Correspondence: justin.ziegler@colostate.edu

**Abstract:** Wildland fire and ecological researchers use empirical and semi-empirical modeling systems to assess fire behavior and danger. This technical note describes the *firebehavioR* package, a porting of two fire behavior modeling systems, Crown Fire Initiation and Spread and a Rothermel-based framework, to the R programming language. We also highlight supporting data objects and functions to predict inputs required for fire behavior estimation. Last, this package contains functions for fifteen indices to express fire danger using weather and/or fuels observations. Specific advantages of predicting fire behavior using R, a free-and-open-source programming language, include freedom to adapt calculations to suit users' needs, transparency of source code, and reduction of workflow inefficiencies, thereby aiding in sophisticated fire behavior analyses.

**Keywords:** Rothermel; surface fire; crown fire; Crown Fire Initiation and Spread; fire modeling; R

## 1. Introduction

Mathematical models of wildland fire behavior are ubiquitous in fire ecology and fire management research. These models use environmental (i.e., meteorological and topographic) and fuels (i.e., dead and live vegetation) descriptors to generate a set of fire behavior predictions. Initially driven by the desire to aid fire management and suppression operations [1], these models are now commonly used to address a variety of research questions related to natural resources and across interdisciplinary fields of scientific inquiry. For example, researchers have used fire behavior models to understand the following: consequences of fuels management [2,3], changes in fire regimes over the Anthropocene [4], wildlife habitat promotion [5], and the role of fire in carbon and water cycles [6,7].

Interactions with mathematical models of fire behavior overwhelmingly occur via software [8]. In North America, the most widely used software employ either the Rothermel system, the Crown Fire Initiation and Spread system (CFIS), or the Canadian Forest Fire Behavior Prediction System (CFFBPS) [9,10]. The Rothermel system is a linkage between the equations of Rothermel [11] which determine spread rate of a surface fire, Van Wagner's [12] model of crown fire initiation, and Rothermel's [13] prediction of crown fire spread rate. The popularity of the Rothermel framework is evident: For example, all simulation-based studies in Fule et al.'s [2] systematic review of fuels treatment efficacy relied on the Rothermel system and this framework is increasingly applied to forested and non-forested ecosystems globally [14]. Despite the popularity of the Rothermel system, assessments of its predictions find that the underlying models in forested ecosystems may underpredict actual fire behavior [15] leading users to expensive calibration efforts in order to generate expected results [9]. As alternatives, using CFFBPS outside of boreal forests is tenuous given as the empirical models within CFFBPS are strictly specific to forest types. Use of CFIS, however, is warranted given the similarity of model inputs with the Rothermel modeling system [9]. And, like the Rothermel modeling systems, inputs are generalized such that any fuels complex can be represented. CFIS links models for crown

fire initiation, active crown fire spread rates, and passive crown fire spread rates [16]. Though CFIS does not describe surface fire, forest managers are often chiefly concerned with diminishing crown fire hazards as crown fires dramatically increase fire behavior [17].

As fire managers heavily use these software applications [8,10], a regular concern is that users may treat fire modeling software as a black-box [18], cautioned even by these models' developers [17,19]. A contributing factor is that compiled, rather than interpreted, software opacify the underlying mathematics [20]. Fortunately, the concern over potential misuse of these modeling systems can be remedied with a functional understanding of these models' mathematics [10,17,21]. Porting fire modeling systems to open-source platforms is one such way to increase transparency of a model's workings [22].

In this technical note, we present the *firebehavioR* package written in the R programming language and illustrate the package's functionality. This package emphasizes fire behavior prediction using the Rothermel and CFIS systems, but also supports relative fire danger estimation using indices. The R software (R Core Team, Vienna, Austria) is commonly used in all scientific disciplines and provides cleansing, wrangling, analysis and visualization of data, permitting streamlined workflow. And, as free-and-open-source software with over 14,000 packages as of May 2019, R provides for complex analyses and flexibility to adapt code for user needs. Released versions of *firebehavioR* with help documentation are available from the Comprehensive R Archive Network [23].

## 2. Functions of *firebehavioR*

### 2.1. The rothermel() Function

*rothermel()* implements the Rothermel modeling system to predict rate of spread (ROS; m/min), fireline intensity (kW/m), and type of fire (surface, passive, or active crown), among other fire behavior metrics (Table 1).

The inputs include selection of standard stylized fuel models, the moisture content of surface fuels, attributes describing moisture and physical attributes of the canopy, and a description of the physical environment (Table 1). Fuel models, stored in *data(fuelMod)*, are parameters for surface fuel categories which yield the fire behavior under a range of environmental conditions characteristic for particular forest or rangeland ecosystems [24,25]. Surface fuel categories include woody particles, classified by their drying time-lag, or diameter, class (1-hr [0–0.635 cm], 10-hr [0.635–2.54 cm], 100-hr [2.54–7.62 cm]), live woody fuel, and live herbaceous fuel. Similar to the software Nexus [26], our implementation of Rothermel [11] also includes litter as a surface fuel class. Users can enter their own fuel moistures or extract moisture scenarios [25] from *data(fuelMoist)*. Users must enter their own crown fuel attributes, though *canFuel()*, an implementation of the Canopy Fuels Stratum Calculator [27], can predict those attributes. Additionally, if the wind adjustment factor (i.e., the ratio of mid-flame wind speed to 20-ft open wind speed) is not known, this can be predicted using *waf()* with known fuelbed characteristics. *rosMult* defaults to 1 to predict average crown fire spread rate but could be altered, such as entering 1.7 to predict the maximum crown fire rate of spread [13]. Lastly, users select a method of determining percent crown fraction burned, a parameter that scales fireline intensity and ROS under passive crown fire conditions (see [26,28,29] for each method's details and [15] for discussion).

### 2.2. The cfis() Function

*cfis()* implements the modeling framework described by Alexander and Cruz [16]. The model estimates likelihood of crown fire initiation [30], and if the likelihood exceeds 0.5, the type of crown fire (active, or passive) is predicted along with crown fire rate of spread (Table 2). An auxiliary output is separation distance; Alexander and Cruz [16] provided an empirical equation to determine how far a spot-fire-generating firebrand must travel in order to escape a primary, moving fire front. The input parameter for this equation is *id*, which is the "elapsed time between a firebrand alighting, subsequent ignition, and the onset of fire spread" [16].

**Table 1.** Inputs and outputs for the *rothermel()* function.

| Input | Description |
|---|---|
| *surfFuel* [1] | Surface fuel attributes consisting of: the fuel model type, either ("s")tatic or ("d")ynamic fuel load transferring; fuel loads (Mg/ha) for litter, 1-hr, 10-hr, 100-hr, live herbaceous, and live woody fuels; surface area-to-volumes ($m^2/m^3$) for litter, 1-hr, 10-hr, 100-hr, live herbaceous, and live woody fuels; fuel bed depth (cm); moisture of extinction (%); and heat content (kJ/kg), in order. |
| *moisture* [1] | Surface fuel moistures on a dry-weight basis (%) for litter, 1-hr 10-hr, 100-hr, live herbaceous, and live woody fuel classes, in order. Entered as $n$ x 6 data frame. |
| *crownFuel* [1] | Canopy fuel attributes consisting of: canopy bulk density ($kg/m^3$); foliar moisture content ("% of dry mass"); canopy base height (m); and canopy fuel load ($kg/m^2$), in order. |
| *enviro* [1] | Environmental variables including: topographic slope ("%"); open windspeed (m/min); wind direction, from uphill (°); and wind adjustment factor (0–1), in order. |
| *rosMult* | Crown fire rate of speed (ROS) multiplier, defaults to 1. Array of length one. |
| *cfbForm* | String specifying estimation method for crown fraction burned. Options are "*sr*" [26], "*w*" [28], or "*f*" [29]. |

| Output [2] | Description |
|---|---|
| *fireBehavior* | Fire behavior summary: fire type, crown fraction burned (%), ROS (m/min), heat per unit area ($kW/m^2$), fireline intensity (kW/m), flame length (m), direction of spread (°), scorch height (m), torching index (m/min), crowning index (m/min), surfacing index (m/min), effective midflame wind speed (m/min), flame residence time (min) |
| *detailSurface* | Surface fire behavior intermediates: potential ROS (m/min), no wind and no slope ROS (m/min), slope factor (-), wind factor (-), characteristic fuel moisture (%), characteristic surface-area-to-volume ratio ($m^2/m^3$), bulk density ($kg/m^3$), packing ratio (-), relative packing ratio (-), reaction intensity ($kW/m^2$), heat source ($kW/m^2$), heat sink ($kJ/m^3$) |
| *detailCrown* | Crown fire behavior intermediates: potential ROS (m/min), no wind & no slope ROS (m/min), slope factor (-), wind factor (-), characteristic fuel moisture (%), characteristic SAV ($m^2/m^3$), bulk density ($kg/m^3$), packing ratio (-), relative packing ratio (-), reaction intensity ($kW/m^2$), heat source ($kW/m^2$), heat sink (kJ/m3) |
| *critInit* | Critical values for crown fire initiation: fireline intensity (kW/m), flame length (m), surface ROS (m/min), canopy base height (m) |
| *critActive* | Critical values for active crown fire: canopy bulk density ($kg/m^3$), crown fire ROS (R'active) (m/min) |
| *critCess* | Critical values for cessation of crown fire: canopy base height (m), cessation index (O') (m/min) |

[1] Inputs entered either as vectors or data frames; [2] output is a list of data frames.

**Table 2.** Inputs and outputs for the *cfis()* function [1].

| Input | Description |
|---|---|
| *fsg* | Fuel stratum gap (m) |
| *u10* | Open wind speed, 10 m above the average canopy height (m/min) |
| *effm* | Estimated fine fuel moisture (%) |
| *sfc* | Surface fuel consumed (Mg/ha) |
| *bbd* | Canopy bulk density ($kg/m^3$) |
| *id* | Ignition delay time for a spotting firebrand (min) |

| Output | Description |
|---|---|
| *type* | Type of fire (surface, passive or active crown fire) |
| *pCrown* | Probability of crown fire (%) |
| *cROS* | Crown fire rate of spread (m/min) |
| *sepDist* | Minimum distance for a spot fire to not be overrun by an advancing fireline (m) |

[1] All inputs are entered either as vectors and output is a data frame.

*2.3. Fire Danger Indices*

Fire danger indices provide a method to gauge relative fire danger based on changing weather and/or fuel conditions. *fireIndex()* calculates static indices, those for which the methods' equations use only instantaneous weather observations. These include the Angstrom, Chandler Burning, Hot-dry-windy, Fuel Moisture, Fosberg Fire Weather, and MacArthur Grassland Mark IV and V indices [31,32]. These all require air temperature, wind speed, and relative humidity, while the MacArthur indices also require the available fuel load and percent grass curing in the case of Mark IV. In contrast, the fire danger indices estimated with *fireIndexKBDI()* are dynamic; these indices are updated daily based on the prior day's value and the current day's conditions. The majority of these indices rely on the Keetch–Byram Drought Index (KBDI) or, the drought factor (DF), a component of KBDI [33]. In addition to KBDI and DF, this function yields the Forest Mark V, the Fosberg Fire Weather Index modified with KBDI, the Fuel Moisture Index modified with KBDI, the Nesterov Index, a modified Nesterov Index, and the Zdenko Index [31–36]. Common inputs to these include air temperature and precipitation amount; additional inputs vary by index and may include mean annual precipitation, wind speed, and relative humidity.

## 3. Demonstration

Fire behavior predictions are demonstrated using plots detailing seven forest stands each measured before and after tree thinning (Table 3). The code to reproduce these results is available in Supplementary Material S1.

**Table 3.** Canopy characteristics of seven forest sites before and after thinning used for fire behavior predictions.

| Site | Status (Pre/Post-Thinning) | Canopy Bulk Density (kg/m$^3$) | Canopy Base Height (m) | Canopy Fuel Load (kg/m$^2$) |
|---|---|---|---|---|
| Heil | pre | 0.14 | 3.63 | 0.837 |
|  | post | 0.10 | 3.69 | 0.628 |
| Kaibab | pre | 0.13 | 4.90 | 2.235 |
|  | post | 0.10 | 6.70 | 1.811 |
| MessG | pre | 0.14 | 3.50 | 1.619 |
|  | post | 0.06 | 3.60 | 0.81 |
| Pike | pre | 0.15 | 2.90 | 2.096 |
|  | post | 0.06 | 3.60 | 0.83 |
| RedF | pre | 0.08 | 2.60 | 0.918 |
|  | post | 0.05 | 3.40 | 0.61 |
| Unc | pre | 0.15 | 4.35 | 2.653 |
|  | post | 0.08 | 4.14 | 1.674 |
| Zuni | pre | 0.09 | 3.83 | 0.914 |
|  | post | 0.03 | 4.60 | 0.396 |

In the first example, using *rothermel()*, we used *data(coForest)* which contains the canopy fuels described in [37–39]. In addition, we selected surface fuel model 10 [24], fuel moisture scenario D1L1 [25], a foliar moisture content of 100%, and a slope of 10% aligned with a midflame wind speed of 16 km/hr. We specified that crown fraction burned was to be calculated as per Scott and Reinhardt [26]. The following results (Figure 1) are a subset of the resulting predictions. The selected results show thinnings yielded an average increase in the torching index of 21%, with the torching index only marginally changed in three sites (Heil, MessG, Unc). Also, on average, the crowning index increased by 63% and the crown fraction burned decreased by 63%.

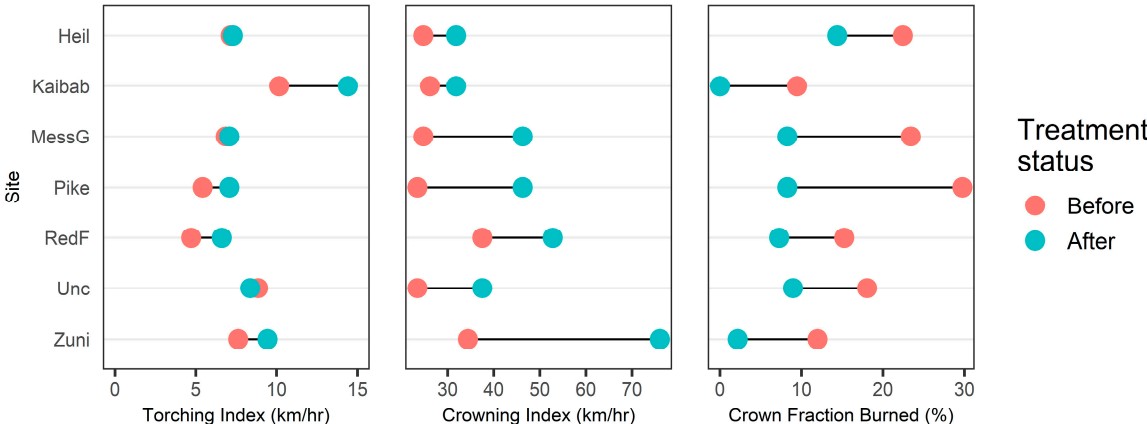

**Figure 1.** Torching index, crowning index and crown fraction burned predicted using the Rothermel modeling system in the *firebehavioR* package.

To demonstrate *cfis()*, we used the same canopy fuels as Table 3. Additional input arguments included the following: a 10-m, open wind speed of 20 km/hr, 0.3 Mg/ha of surface fuel load consumed, and an estimated fine fuel moisture of 8%. Figure 2 shows thinning had the most impact on the Kaibab and Zuni sites, mirroring relative performance as predicted by the torching index using the Rothermel modeling system (Figure 1). On four sites (Heil, MessG, Pike, Unc), thinnings had similar effects, reducing crown fire rates of spread by approximately 50% to 60% (Figure 2) as the fire type dropped from active to passive. At Kaibab, thinnings reduced, the crowning probability dropped below 50%, eliminating any crown fire. At RedF and Zuni, where fire type persisted as passive crown fire despite thinning, crown fire rate of spread increased.

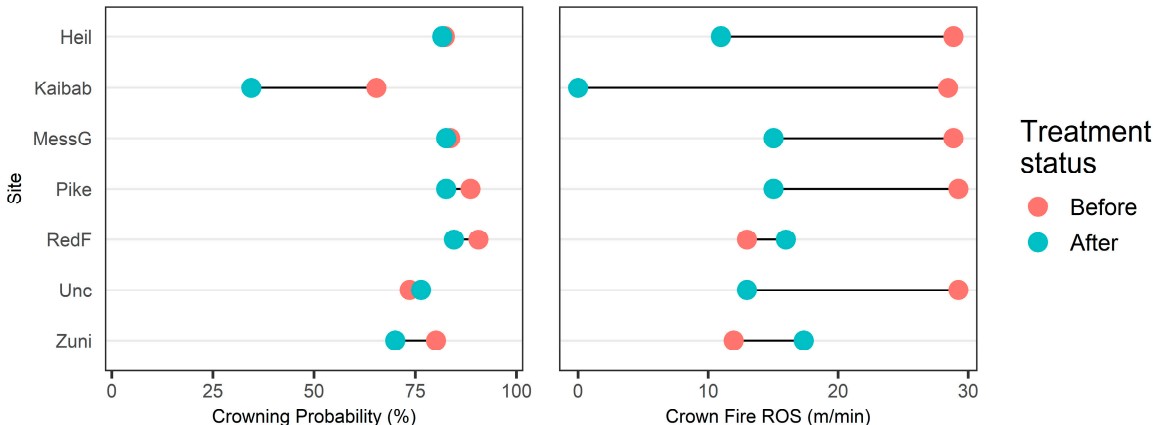

**Figure 2.** Probability of crowning and predicted crown fire rate of spread (ROS) predicted using the Crown Fire Initiation and Spread model in the *firebehavioR* package.

## 4. Discussion

The primary purpose of *firebehavioR* was to compute fire behavior characteristics using either the linked sub models of the Rothermel modeling system or CFIS. A vigthinette with further demonstrability is available in Supplementary Material S2. While CFIS is applicable only for forested scenarios, Supplementary Material S2 also illustrates how canopy fuels can be omitted to simulate fire in rangelands using the Rothermel modeling system. Further, Supplementary Material S2 contains a table with all parameters and their respective measurement units; when choice of parameter values is unclear, users are encouraged to review cited source literature of fire models and danger indices, or references such as the National Wildfire Coordinating Group (NWCG) [40].

To our knowledge, no other reposited R package has comparable functionality. Of the most similar packages, the *Rothermel* package [20] predicts Rothermel's [11] surface fire spread rate which, while applicable for many ecosystems, may be limiting where crown fire initiation or spread is possible. The *Medfate* package [41] also predicts surface fire spread rates and indices of crown fire hazard though based on the substantial modifications following Sandberg et al. [42]. No package in R has implemented the CFIS modeling system. Our package joins a growing ecosystem of wildland fire-related R packages. For example, *cffdrs* [43] has implemented the Canadian Forest Fire Danger Rating System, *Caliver* [44] can be used to assess fire danger of spatially gridded data, and *PWFSLsmoke* [45] is available to analyze particulate emissions. By offering our package to a small but growing collection of fire-related packages, we give users greater flexibility to suit specific informational needs.

We verified source code by performing multiple tests comparing intermediate and final results against other software. Tests showed perfect agreement with CFIS. Compared against software deploying the Rothermel modeling system [14,26], there were minor discrepancies which came from two sources. Firstly, like many other implementations of this modeling system, we converted from SI to United States customary unit systems for calculations and back to SI; the precision of conversion factors will affect solutions. Second, different software employ different modifications, such as the limit placed on the wind factor used in predicting surface fire rate of spread in BehavePlus [14], the pooling of live woody and herbaceous fuel moistures in Nexus [26], or the specific method to calculate crown fraction burned. Of these, the former error source is minimal and the latter may be substantial [9]. This speaks to the importance of citing research software and specifying parameterization.

Compared to other software deploying the same fire modeling systems, the implementation in R is advantageous for two reasons: source code is free-and-open-source and, since users can interface via a read-evaluate-print-loop console, the interactivity and adaptability facilitates teaching and research. While explanations of fire behavior model mechanics are frequently reiterated in the scientific literature (e.g., [14]), for some learners, the ability to read and interact with algorithms can lead to greater conceptual understanding than generalized equations presented in literature [22]. Our implementation of fire modeling systems also increases transparency of fire behavior calculations; this may hold value for interdisciplinary research efforts wherein expertise lies outside fire science. It is worth noting that lack of a graphical user interface, some requisite programming ability, and specific knowledge of R syntax means *firebehavioR* cannot satisfy all users.

Further, by linking inputs and outputs across packages, users can efficiently conduct complex analyses; this is because data are always stored as structured data types. This reduces data wrangling during analytical procedures [22]. Since users can seamlessly integrate *firebehavioR* inputs and outputs to packages containing climatological, topographic, vegetative, geographic, and other models this facilitates sophisticated modeling of dynamic pyro-ecological systems. For example, the two aforementioned R packages illustrate the advantage of interoperability. The *Rothermel* package [20] includes a function with a dependency on a genetic algorithms package in order to fit custom fuel model parameters based on observed fire behavior. In addition, *Medfate* [41] links models of fire behavior and severity to ecophysiological models to simulate feedbacks between fire and vegetation dynamics.

In conclusion, *firebehavioR* provides an additional open-source tool to move fire science and related ecological study forward. Contributions are welcome as development on *firebehavioR* continues.

## 5. Patents

The firebehavioR package for the R programming language is distributed under the GNU General Public License v2.0.

**Supplementary Materials:** The following are available online at http://www.mdpi.com/2571-6255/2/3/41/s1, Supplementary S1: Reproducible demonstration code; Supplementary S2: An Introduction to the firebehavioR package.

**Author Contributions:** Conceptualization, J.P.Z.; methodology, J.P.Z, C.M.H., W.M.; software, J.P.Z.; validation, J.P.Z., C.M.H.; resources, J.P.Z.; data curation, J.P.Z.; writing—original draft preparation, J.P.Z.; writing—review and editing, J.P.Z., C.M.H., W.M.; visualization, J.P.Z.; supervision, C.M.H.; funding acquisition, C.M.H., W.M.

**Funding:** This research was funded by the Joint Fire Science Program, project 14-1-01-18 "Assessing factors that influence landscape fuels treatment effectiveness".

**Acknowledgments:** Scott Ritter of the Department of Forest and Rangeland Resources, Colorado State University was a beta tester. This work was enabled by concurrent efforts to implement fire behavior predictions into the Wildland-urban interface Fire Dynamics Simulator (WFDS).

**Conflicts of Interest:** The author declares no conflicts of interest.

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
