# Peer review of "firebehavioR: An R Package for Fire Behavior and Danger Analysis"

_fire, doi:10.3390/fire2030041_

Round 1

Reviewer 1 Report

I found the manuscript by Ziegler and colleagues well written and concise. It describes a new package in the R programming language including several fire models that are established and used by practitioners and researchers in fire science. The R package is accessible on the CRAN repository, and it can be downloaded together with the user guide according to R standards. I run few simulations to test for its ergonomics and conformity to original models predictions: the few tests I run showed equivalence, except for very little differences (not relevant) in the Rothermel’s surface model, due to conversion factors to the international metric system, as acknowledged in the discussion section. Consequently mi overall assessment of the manuscript and R package is positive and my opinion it is that the manuscript should be considered for publication by the editor after few minor revisions listed here after that I would like to be addressed.

Minor comments

In my opinion, the manuscript should acknowledge a number of previous work on the implementation of fire models in the R programming language, either in introduction section or in the discussion. This can add value to the manuscript, since there is not a state of art of this promising interaction between R and fire science. In particular, I refer to the following studies:

the package “fwi.fwp” (available at: https://cran.r-project.org/web/packages/fwi.fbp/index.html) which calculates the outputs of the two main components of the Canadian Forest Fire Danger Rating System and a recent update by the same authors with  the package ‘cffdrs’ (https://cran.r-project.org/web/packages/cffdrs/cffdrs.pdf);

the Caliver package to analyze fire gridded data, see: https://github.com/ecmwf/caliver described in the publication of Vitolo et al. (2018);

the PWFSLsmoke package (https://cran.r-project.org/web/packages/PWFSLSmoke/index.html) which includes utilities for working with air quality monitoring data generated by wildfire.

But see also the comment to LN149.

LN40-41: in my opinion, the sentence “leading users to alter inputs in order to generate expected results” is not fully correct, particularly as regards the Rothermel (1972) surface fire spread model. Indeed, the Rothermel surface fire spread model requires a calibration process against observed rate of spread, which is made by either stylized standard fuel models selection or by building custom fuel models. Consequently, I suggest to change the sentence in “leading users to expensive calibration procedures in order to generate expected results”.

LN58-59 and 100: to my knowledge the term “fire hazard” is not appropriate to refer to fire weather indices. Indeed “fire hazard” is commonly used to refer to potential intense fire behavior as a consequence of hazardous fuel accumulation. Se, as example, how fire hazard is used in the paper from Fernandes & Botelho 2003. I believe the term “fire danger” is more appropriate for model intended to predict effect of weather on the fuel moisture status (see as example Wastl et al. 2012).

LN149: change the reference [21] in [20]

LN149: in referring to the “Rothermel” package, it would be fair to acknowledge two functions that distinguish this package from what implemented in the “firebehavioR” package: the bestFM function and the gaRoth function (Ascoli et al. 2015) which allow to select the best standard fuel model and calibrate new fuel models against rate of spread observations.

References

Ascoli, D., Vacchiano, G., Motta, R., & Bovio, G. (2015). Building Rothermel fire behaviour fuel models by genetic algorithm optimisation. International Journal of Wildland Fire, 24(3), 317-328.

Fernandes, P. M., & Botelho, H. S. (2003). A review of prescribed burning effectiveness in fire hazard reduction. International Journal of wildland fire12(2), 117-128.

Vitolo, C., Di Giuseppe, F., & D’Andrea, M. (2018). Caliver: An R package for CALIbration and VERification of forest fire gridded model outputs. PloS one, 13(1), e0189419.

Wastl, C., Schunk, C., Leuchner, M., Pezzatti, G. B., & Menzel, A. (2012). Recent climate change: long-term trends in meteorological forest fire danger in the Alps. Agricultural and Forest Meteorology, 162, 1-13.

Author Response

Reviewer: I found the manuscript by Ziegler and colleagues well written and concise. It describes a new package in the R programming language including several fire models that are established and used by practitioners and researchers in fire science. The R package is accessible on the CRAN repository, and it can be downloaded together with the user guide according to R standards. I run few simulations to test for its ergonomics and conformity to original models predictions: the few tests I run showed equivalence, except for very little differences (not relevant) in the Rothermel’s surface model, due to conversion factors to the international metric system, as acknowledged in the discussion section. Consequently mi overall assessment of the manuscript and R package is positive and my opinion it is that the manuscript should be considered for publication by the editor after few minor revisions listed here after that I would like to be addressed.

Authors: Thank you Reviewer for taking the time and effort to judge our manuscript for publication. We expect that the revisions in response to your comments ultimately provide a better product for readers.

In addition, it is our estimation that fire researchers often develop, on an ad-hoc basis, specific research software where and when current solutions are lacking.  The fire research community may benefit from a review and assessment of current available packages in order to guide future development and increased coverage of fire research needs. As the reviewer appears intimate with the process of package development for fire research, the reviewer is welcome to contact the lead author for further discussion.

Reviewer:

In my opinion, the manuscript should acknowledge a number of previous work on the implementation of fire models in the R programming language, either in introduction section or in the discussion. This can add value to the manuscript, since there is not a state of art of this promising interaction between R and fire science. In particular, I refer to the following studies:

the package “fwi.fwp” (available at: https://cran.r-project.org/web/packages/fwi.fbp/index.html) which calculates the outputs of the two main components of the Canadian Forest Fire Danger Rating System and a recent update by the same authors with  the package ‘cffdrs’ (https://cran.r-project.org/web/packages/cffdrs/cffdrs.pdf);

the Caliver package to analyze fire gridded data, see: https://github.com/ecmwf/caliver described in the publication of Vitolo et al. (2018);

the PWFSLsmoke package (https://cran.r-project.org/web/packages/PWFSLSmoke/index.html) which includes utilities for working with air quality monitoring data generated by wildfire.

But see also the comment to LN149.

Author: The Reviewer suggests that we provide greater background on fire-related packages implemented in the R language. In general, we feel that discussing the greater fire-related R ecosystem is slightly extraneous and not relevant for our specific motivation (the Introduction), but mention of other packages is worthy for placing our package in context. To that end we have added the following in our discussion (LN164): “Our package joins a growing ecosystem of wildland fire-related R packages. For example, cffdrs [43] has implemented the Canadian Forest Fire Danger Rating System, Caliver [44] can be used to assess fire danger of spatially gridded data, and PWFSLsmoke [45] is available to analyze particulate emissions. By offering our package to a small but growing collection of fire-related packages, we give users greater flexibility to suit specific informational needs.”

As an aside, we would be happy to collaborate on a review of current fire research packages in the open-source community in order to highlight current areas of coverage and research software development needs.

Reviewer: LN40-41: in my opinion, the sentence “leading users to alter inputs in order to generate expected results” is not fully correct, particularly as regards the Rothermel (1972) surface fire spread model. Indeed, the Rothermel surface fire spread model requires a calibration process against observed rate of spread, which is made by either stylized standard fuel models selection or by building custom fuel models. Consequently, I suggest to change the sentence in “leading users to expensive calibration procedures in order to generate expected results”.

Authors: We agree that a deliberate process of selecting custom fuel model parameters is one option to generate expected results, but often environmental inputs are also amplified on an ad-hoc iterative basis until results are satisfactory. As stated by our source (Scott 2006) and by our experience working for or with land management agencies, it is common to use standard fuel models but amplify the winds or lower the canopy base height in order to generate expected results. This is especially true in instances where rate of spreads and fireline intensities are as expected but results suggest an underprediction in crown fire potential. Regardless, we can agree that your suggestion all constitute forms of time-consuming calibration based on prior experience or measurements; thus, we have made the suggested revision on LN41.

Reviewer: LN58-59 and 100: to my knowledge the term “fire hazard” is not appropriate to refer to fire weather indices. Indeed “fire hazard” is commonly used to refer to potential intense fire behavior as a consequence of hazardous fuel accumulation. Se, as example, how fire hazard is used in the paper from Fernandes & Botelho 2003. I believe the term “fire danger” is more appropriate for model intended to predict effect of weather on the fuel moisture status (see as example Wastl et al. 2012).

Authors: The reviewer brings up an important terminological distinction to which we have we have corrected by changing hazard to danger throughout the manuscript and in Appendix B. Thank you.

Reviewer: LN149: change the reference [21] in [20]

Authors: Thank you for pointing out the correct reference.

Reviewer: LN149: in referring to the “Rothermel” package, it would be fair to acknowledge two functions that distinguish this package from what implemented in the “firebehavioR” package: the bestFM function and the gaRoth function (Ascoli et al. 2015) which allow to select the best standard fuel model and calibrate new fuel models against rate of spread observations.

Authors: We appreciate the functional differences between various fire-related R packages. For clarity, we retain the focus of this paragraph as what distinguishes our package from others. Here we have  added (LN159) a distinction which makes clear how the Rothermel package is similar and dissimilar to our package: “Of the most similar packages, the Rothermel package [20] predicts Rothermel’s [11] surface fire spread rate which, while applicable for many ecosystems, may be limiting where crown fire initiation or spread is possible.”

However, your comment speaks to the advantage of being able to conduct analyses in a single software platform rather than disparate software. On LN193, we have added: “For example, the two aforementioned R packages illustrate the advantage of interoperability. The Rothermel package [20] includes a function with a dependency on a genetic algorithms package to fit custom fuel model parameters based on observed fire behavior. And, Medfate [31] links models of fire behavior and severity to ecophysiological models to simulate feedbacks between fire and vegetation dynamics.”.

Reviewer 2 Report

This technical note describes an R package for Fire Behavior analysis that was recently released.  The text provides a good justification for the formation of the package and how it connects to other available computational sources. I have only a few minor comments:

L31 - Overwhelming occur- should this be overwhelmingly?

L49 - Is the intent for this package to be solely used by Forest managers?  In other places I get the impression that the authors consider this package applicable to grasslands as well.  Some clarification on the utility across the forest-shrubland-grassland gradient would be beneficial. L74 uses the word rangeland - but I'd think grassland would be better?

Table 1 - surfFuel - moisture of extinction how does this differ from fuel moisture.  It's really the only term that might need some more description

Author Response

Reviewer: This technical note describes an R package for Fire Behavior analysis that was recently released.  The text provides a good justification for the formation of the package and how it connects to other available computational sources. I have only a few minor comments:

Authors: We graciously thank the reviewer for their time and effort in judging the manuscript and making suggestions for improvement. Though not every comment was revised in complete accordance with suggestions, we believe our revisions will satisfy the reviewer and readers.

Reviewer: L31 - Overwhelming occur- should this be overwhelmingly?

Authors: Noted and changed in the manuscript. Thank you.

Reviewer: L49 - Is the intent for this package to be solely used by Forest managers?  In other places I get the impression that the authors consider this package applicable to grasslands as well.  Some clarification on the utility across the forest-shrubland-grassland gradient would be beneficial. L74 uses the word rangeland - but I'd think grassland would be better?

Authors: It is not our intent to delineate fire model use by land cover types, specifically because, with exceptions of forest canopy parameters, the Rothermel framework does not discriminate across land cover type. Our intent is to provide the tools with the appropriate flexibility for users specific cases, wherever they might be (for example, selection or customization of fuel models within the Rothermel fire modelling framework). We provided clarification on L152 which now states: “While CFIS is applicable only for forested scenarios, Appendix B also illustrates how canopy fuels can be omitted to simulate fire in rangelands using the Rothermel modeling system.”

In addition, some of our language erroneously was exclusive of rangelands (chiefly shrublands, grasslands and tundra). We corrected for this in various locations. For example, we clarified: in L40, “and this framework is increasingly applied to forested and non-forested ecosystems globally “; in L42, “assessments of its predictions find that the underlying models in forested ecosystems may underpredict actual fire hazard” in order to clarify that CFIS is an alternative specifically for forestlands. And in L52, we revised our statement: “As forest fire managers heavily use these software applications…”.

We have also clarified by noting that our package is distinct from the Rothermel package in that surface fire behavior prediction is adequate for non-forested scenarios whereas our package adds the capacity to account for canopy fuel hazard (L158): “Of the most similar packages, the Rothermel package [20] predicts Rothermel’s [11] surface fire spread rate which, while applicable for many ecosystems, may be limiting where crown fire initiation or spread is possible.”

Reviewer: Table 1 - surfFuel - moisture of extinction how does this differ from fuel moisture.  It's really the only term that might need some more description

Authors: We feel that there is a great deal of terminology, often with model-specific definitions, which require further description in order to responsibly use (e.g., relative packing ratio) and whose terms are less self-evident that the moisture of extinction. What terms are familiar with users is likely to depend on their knowledge and experience with fire models. While the onus is on the user to responsibly choose which fire model to use (and how to responsibly use models given their understanding) and how to choose parameter values (like most technical software), we did add a statement (L154) to advise users: “Further, Appendix B contains a table with all parameters and their respective measurement units; when choice of parameter values is unclear, users are encouraged to review cited source literature of fire models and danger indices, or references such as NWCG [40].”
